# Cecropin A Alleviates LPS-Induced Oxidative Stress and Apoptosis of Bovine Endometrial Epithelial Cells

**DOI:** 10.3390/ani14050768

**Published:** 2024-02-29

**Authors:** Yu Zhao, Yang Zhang, Mingkun Sun, Bowen Li, Yuqiong Li, Song Hua

**Affiliations:** 1College of Veterinary Medicine, Northwest A&F University, Yangling 712100, China; zhaoyu1999@nwafu.edu.cn (Y.Z.); zhangy0523@126.com (Y.Z.); sunmingkun@nwafu.edu.cn (M.S.); libowen_0825@nwafu.edu.cn (B.L.); 2Mianyang Habio Bioengineering Co., Ltd., Mianyang 621000, China; 3Laboratory Institute of Animal Science, Ningxia Academy of Agricultural and Forestry Sciences, Yinchuan 750000, China; nxlyq1973@163.com

**Keywords:** cecropin A, lipopolysaccharide, oxidative stress, inflammation, apoptosis, endometrial epithelial cells, dairy cows

## Abstract

**Simple Summary:**

Dairy cows receiving a prolonged high-concentrate diet are highly susceptible to sub-acute ruminal acidosis, leading to an elevated concentration of lipopolysaccharides (LPSs) in the peripheral blood circulation. A high concentration of LPS can cause severe oxidative stress, leading to various inflammations, which can cause inflammatory damage and even apoptosis. Antimicrobial peptides (AMPs) not only have antimicrobial effects but also play roles in cells including anti-inflammation, antioxidative stress, and anti-apoptosis. In this study, we found that cecropin A, as an AMP, relieved LPS-induced oxidative stress in dairy cow endometrial epithelial cells and inhibited the inflammatory response caused by oxidative damage. Additionally, we discovered that cecropin A can effectively alleviate LPS-induced cell apoptosis by inhibiting the mitochondrial-dependent apoptotic pathway.

**Abstract:**

Dairy cows receiving a prolonged high-concentrate diet express an elevated concentration of lipopolysaccharides (LPSs) in the peripheral blood circulation, accompanied by a series of systemic inflammatory responses; however, the specific impacts of inflammation are yet to be determined. Cecropin-like antimicrobial peptides have become a research hotspot regarding antimicrobial peptides because of their excellent anti-inflammatory activities, and cecropin A is a major member of the cecropin family. To elucidate the mechanism of cecropin A as anti-inflammatory under the condition of sub-acute ruminal acidosis (SARA) in dairy cows, we induced inflammation in bEECs with LPS (10 µg/mL) and then added cecropin A (25 µM). Afterwards, we detected three categories of indexes including oxidative stress indices, inflammation-related genes, and apoptosis-related genes in bovine endometrial epithelial cells (bEECs). The results indicated that cecropin A has the ability to reduce inflammatory factors TNF-α, IL-1β, and IL-8 and inhibit the MAPK pathway to alleviate inflammation. In addition, cecropin A is able to reduce reactive oxygen species (ROS) levels and alleviates LPS-induced oxidative stress and mitochondrial dysfunction by downregulating NADPH Oxidase (NOX), and upregulating catalase (CAT), glutathione peroxidase (GPX), and superoxide dismutase (SOD). Furthermore, cecropin A demonstrates the ability to inhibit apoptosis by suppressing the mitochondrial-dependent apoptotic pathway, specifically Fas/FasL-caspase-8/-3. The observed increase in the Bcl-2/Bax ratio, a known apoptosis regulator, further supports this finding. In conclusion, our study presents novel solutions for addressing inflammatory responses associated with SARA.

## 1. Introduction

Long-term feeding of dairy cows with a high-concentrate diet poses a significant risk of sub-acute ruminal acidosis (SARA), leading to a high concentration of lipopolysaccharides (LPSs) in the circulatory system and triggering a systemic inflammatory response [1]. A previous study demonstrated that elevated LPS levels correlate with increased reactive oxygen species (ROS) and reduced mitochondrial membrane potential, ultimately causing mitochondrial dysfunction [2]. Furthermore, LPS leads to the upregulation of NADPH Oxidase (NOX), then resulting in an increase in ROS levels, which is a marker of oxidative stress [3]. The increased ROS levels are able to exacerbate mitochondrial dysfunction, diminish mitochondrial membrane potential, and activate cell apoptosis [4]. Additionally, high ROS concentrations are pivotal indices in numerous inflammatory diseases, and oxidative stress is a main source of inflammation [5].

Oxidative stress can prompt the release of various pro-inflammatory cytokines, including Interleukin-1β (IL-1β) and tumor necrosis factor α (TNF-α). These cytokines play a role in promoting apoptotic signaling transduction induced by the Fas ligand (FasL) [6]. TNF-α, a crucial player in cell apoptosis, activates the Fas/FasL-caspase-8/-3 pathway, leading to increased Caspase-3 activity and apoptosis [7]. Moreover, IL-1β activates the Fas/FasL-caspase-8/-3 pathway by stimulating Bid and JNK1/2, initiating a caspase cascade reaction that culminates in cell apoptosis [6].

The Fas/FasL-caspase-8/-3 pathway is a significant mitochondrial-dependent apoptotic pathway. Fas is a transmembrane protein belonging to the tumor necrosis factor receptor superfamily, possessing a unique death domain. Upon binding to death ligands like FasL or TNF-α, the death domains of three Fas molecules cluster together. This clustering attracts another cytoplasmic protein, Fas-associated death domain protein (FADD), carrying a death domain [8,9]. FADD, with a death effector domain (DED), forms an apoptotic complex by binding to the precursor of Caspase-8 through homodimerization. Caspase-8, also possessing a DED, undergoes hydrolysis and activation upon binding to FADD. Activated Caspase-8 triggers the conversion of Bid into tBid, a member of the BH3 protein family [10]. Subsequently, tBid translocates to the mitochondrial membrane, activating pro-apoptotic factors from the Bcl-2 family and inhibiting anti-apoptotic factors from the same family. Simultaneously, this stimulates the formation of the BAK-BAX polymer outside the mitochondrial membrane, releasing cytochrome C from mitochondria into the cytoplasm. This event triggers the activation of Caspase-9 through the mitochondrial apoptotic pathway [11,12]. As Caspase-9 promotes the activation of Caspase-3, a cascade of caspase reactions ensues, culminating in cell apoptosis [13,14] (see Figure 1).

Cecropin A, an antimicrobial peptide (AMP) comprising 37 amino acid residues, is distinguished by its anti-inflammatory activity [15,16]. Despite its common use in production as a feed additive to enhance metabolism and immunity by regulating gastrointestinal microbiota, a substantial body of literature supports the direct inhibition of LPS-induced inflammatory factors and pathways by cecropin A. Previous studies have highlighted its capability to inhibit LPS binding to Cluster of Differentiation 14 (CD14), thus preventing LPS-induced TNF-α production in macrophages [17]. Other research has shown that cecropin A exerts anti-inflammatory and anti-apoptotic effects by inhibiting the expression of cytokines such as TNF-α, IL-1β, and MIP-2 [18]. Additionally, cecropin A exhibits potent anti-apoptotic activity by inhibiting the JNK and p38-MAPK signaling pathways and suppressing the expression of COX-2 [19,20].

Furthermore, in our observations on pastures, supplementing dairy cows with cecropin A as a feed additive in the context of a high-concentrate diet significantly suppressed inflammatory factor levels in the peripheral blood circulation. This led us to infer that the overall reduction in the reproductive rate of cows could be associated with the inflammatory response induced by high-concentrate feeding, and cecropin A demonstrated an inhibitory effect on this process. In this study, we delved into the detailed mechanism by which cecropin A alleviates LPS-induced oxidative stress, inflammation, and apoptosis in bovine endometrial epithelial cells (bEECs). We confirmed that cecropin A can alleviate LPS-induced oxidative stress, inflammation, and cell apoptosis through a mitochondrial-dependent pathway, showcasing its significant potential in the treatment of various inflammatory diseases. This discovery opens a new avenue for addressing endometritis and offers insights into tackling the reduced embryo implantation rate attributed to high-concentrate feeding.

## 2. Materials and Methods

### 2.1. Tissue Collection

In our study, all uterine samples were collected from Holstein cows (an average weight of 610 kg, 1–2 parity) fed with a 100% total mixed ration (TMR) at 8:00 and 19:00 every day for 1 year, meeting the revised nutritional requirements (NRC) of the US National Research Council, and not suffering from mammitis, hoof disease, or other diseases determined based on a veterinary clinical diagnosis. The ingredients and nutrient contents are shown in Table 1. The uterine samples were sourced from Holstein cows with SARA, detected using a previously described method [21], receiving a high-concentrate diet (concentrate/crude = 5:5). Using Holstein cows as controls, they received a high-concentrate diet supplemented with 0.012% of cecropin A (Q/DXN 063-2021, Viteling Antibiotic-Free Breeding Technology, Xi’an, China) as a feed additive mixed into the TMR. Four cows from each of the two feeding regimes were selected for slaughter and sampled for the uterine horn. Fresh uterus horn specimens were procured from the slaughterhouse and promptly transported to a laboratory, maintaining a two-hour window using ice. Tissue slices measuring 4 × 4 mm^2^ were obtained using biopsy forceps. The collected tissues were submerged in a saline solution (0.9%). Subsequently, the uterine tissue samples underwent a bifurcation process: one portion was immediately immersed in liquid nitrogen for RNA extraction, while the other segment was fixed in 4% polyformaldehyde and subjected to conventional hematoxylin and eosin (HE) staining for histopathological examination. Additionally, immunohistochemical (IHC) staining was conducted to assess the apoptotic status of the cells. 

The ratio of the concentrate to forage is 5:5.The premix contained 10,000 IU/kg of vitamin A, 5500 IU/kg of vitamin D3, 2000 IU/kg of vitamin E, and 2200 IU/kg of vitamin K3. The diet contained 61.50 mg/kg of zinc, 56.50 mg/kg of manganese, 70.50 mg/kg of iron, 10.75 mg/kg of copper, 0.75 mg/kg of iodine, 0.45 mg/kg of cobalt, and 0.50 mg/kg of selenium.Cows as controls were fed with a high-concentrate diet supplemented with cecropin A at a rate of 0.012% of the concentrate. The cows we sampled tissues from who were detected with SARA were fed with a high-concentrate diet with no cecropin A.

### 2.2. Cell Treatment

Bovine endometrial epithelial cells (bEECs), from a cell line maintained in our laboratory [22], were thawed and cultured in 6-well plates in a medium comprising 90% high-glucose Dulbecco’s modified Eagle’s medium (DMEM, BL304A, Biosharp, Guangzhou, China) and 10% fetal bovine serum (FBS, Z7186FBS-500, ZETA LIFE, San Francisco, CA, USA) with 37 °C in a 5% CO_2_ and atmospheric air environment with saturation humidity. To determine the optimal time and concentration of cecropin A, we conducted initial experiments using the CCK-8 assay. Cecropin A (a chemical reagent, HY-P1539, MedChemExpress, Monmouth Junction, NJ, USA) was tested at concentrations of 0, 12.5, 25, and 50 μM across four concentration groups, and exposure times of 0, 6, 12, and 24 h were evaluated at four time points to assess cell viability and determine the optimal concentration and treatment duration.

Subsequently, once the cells reached 70–80% confluence, they were exposed to 10 μg/mL of lipopolysaccharides (LPSs, L4391, Sigma, Saint Louis, MO, USA), incubated for 24 h, and used as the LPS-treated group. Cells initially exposed to LPS (10 μg/mL) for 6 h and subsequently supplemented with 25 μM of cecropin A for an additional 18 h were used as the cecropin A + LPS-treated group. Those cells treated neither with LPS nor cecropin A were used as the control group.

### 2.3. Quantitative Reverse Transcription PCR

Total cellular RNA was extracted from bEECs utilizing the AG RNAex Pro Reagent (AG11711, Accurate Biology, Changsha, China), and the concentration and purity of RNA were evaluated using a NanoDrop 2000 spectrophotometer (Thermo Science, Waltham, MA, USA). Subsequently, 1 μg of RNA underwent a genomic DNA elimination reaction, followed by reverse transcription into a cDNA template utilizing the Evo M-MLV RT Kit with gDNA Clean for qPCR II (AG11711, Accurate Biology, Changsha, China).

Quantitative PCR was conducted on a LightCycler^®^480 system (Roche, Basel, Switzerland) using the ChamQ SYBR qPCR Master Mix (Q311-02, Vazyme, Nanjing, China). The reaction conditions were as follows: an initial denaturation at 95 °C for 2 min, followed by 40 cycles of denaturation at 95 °C for 10 s, and annealing/extension at 60 °C for 30 s. The β-actin gene served as the reference gene, and statistical comparisons were executed using the 2−ΔΔCt method. A previously described method [23] was used to calculate the Primer Efficiencies. Detailed primer sequences for each gene are provided in Table 2.

### 2.4. Western Blotting

Total proteins were extracted from bEECs utilizing a protein extraction kit (KeyGEN, Changchun, China), and the protein concentration was determined using a BCA Protein Assay Kit (KGPBCA, KeyGEN BioTECH, Shanghai, China). Subsequently, the proteins, separated using SDS-PAGE (15%), were transferred to a PVDF membrane.

To block the membrane, it was shaken in 10% skimmed milk powder in TBST (50 mmol/L of Tris, pH 7.6, 150 mmol/L of NaCl, and 0.1% Tween 20) for 2 h at room temperature. The membrane was then incubated for 12 h at 4 °C with specific primary antibodies, including mouse anti-SOD2 (1:1000, SC-137254, Santa Cruz Biotechnology Inc., Santa Cruz, CA, USA), mouse anti-p38MAPK alpha (1:1000, SC-81621, Santa Cruz Biotechnology Inc., Santa Cruz, CA, USA), rabbit anti-FADD (1:1000, DF7674, Affinity Biosciences, Beijing, China), rabbit anti-caspase-8 (1:1000, AF6442, Affinity Biosciences, Beijing, China), rabbit anti-cleaved-caspase-3 (1:1000, AF7022, Affinity Biosciences, Beijing, China), rabbit anti-cytochrome C (1:1000, AF0146, Affinity Biosciences, Beijing, China), rabbit anti-BAX (1:1000, AF0120, Affinity Biosciences, Beijing, China), and rabbit anti-Bcl-2 (1:1000, 12789-1-AP, Proteintech, Rosemont, IL, USA). Following this, the membrane went through incubation with secondary antibodies, including Goat Anti-Rabbit IgG (H + L) (S0001, Affinity Biosciences, Beijing, China) or Goat Anti-Mouse IgG (H + L) (S0002, Affinity Biosciences, Beijing, China), for 2 h at room temperature.

After washing three times with TBST, the protein bands were visualized using an ECL Super Sensitive Kit (DiNing, Beijing, China) through a chemiluminescent detection system. β-Actin served as an internal reference, and the intensity analysis of protein bands was conducted using ImageJ 1.47 software.

### 2.5. Detection of Intracellular ROS

Intracellular reactive oxygen species (ROS) levels were determined using 2′,7′-dichlorofluorescein diacetate (DCFH-DA) according to the manufacturer’s instructions (S0033S, Beyotime, Shanghai, China). Briefly, when bEECs cultured in 12-well plates reached 70–80% cell density, the medium was replaced with a serum-free medium containing 10 μmol/L of DCFH-DA, and the cells were incubated at a temperature of 37 °C for half an hour. Meanwhile, the Rosup provided by the kit served as a positive control for inducing oxidative stress. All samples were visualized under laser scanning microscopy (Nikon, Tokyo, Japan), and the fluorescence intensity was quantified using ImageJ 1.47 software (National Institutes of Health, Bethesda, MD, USA).

### 2.6. Mitochondrial Membrane Potential Assay

The JC-1 kit (C2006, Beyotime, Shanghai, China) was employed to assess the mitochondrial membrane potential (MMP) of bEECs. In this assay, JC-1 exhibits red fluorescence at high MMP values, while it displays green fluorescence at low MMP values. The cell culture medium was aspirated, and cells were incubated in a fresh medium containing JC-1 (0.6 mL) for 20 min at 37 °C. Afterward, the fluorescence intensity of bEECs was examined using laser scanning microscopy following three washes with PBS.

### 2.7. Flow Cytometry

The flow cytometry analysis was conducted using the BD FACSAria™ III system (USA) along with the Annexin V-FITC/PI Apoptosis Detection Kit (KGA105-KGA108, KeyGEN BioTECH, Shanghai, China) to determine the apoptosis rate of bEECs subjected to various treatments. The bEECs were exposed to distinct treatment groups and subsequently analyzed with flow cytometry (BD FACSAria™ III, Piscataway, NJ, USA), employing the Annexin V-FITC/PI Apoptosis Detection Kit (KGA105-KGA108, KeyGEN BioTECH, Shanghai, China). All data acquired from the flow cytometry analysis were processed using FlowJo V10 software (Treestar, San Francisco, CA, USA).

### 2.8. Cellular Immunofluorescence

The bEECs underwent fixation with 4% paraformaldehyde for 30 min, followed by permeabilization using 0.2% Triton-X100 and blocking with 1.0% BSA at room temperature for 1 h. Subsequently, the cells were coincubated with specific primary antibodies at 4 °C for 12 h. After washing with PBS to remove the specific primary antibodies, the cells were coincubated with fluorescent secondary antibodies at room temperature for 2 h. The slides were then sealed in a sealing solution with an anti-fluorescence quencher and DAPI (P0131, Beyotime, Shanghai, China) before being observed using a laser confocal microscope (Nikon, Tokyo, Japan). The fluorescence intensity was analyzed using ImageJ 1.47 software.

### 2.9. Statistics

All values were derived from a minimum of three independent experiments for each condition in the current study. The data were expressed as means ± SD and subjected to the statistical analysis using SPSS (IBM SPSS Statistics 20, Chicago, IL, USA). The significant difference in mean values was assessed through a one-way analysis of variance (ANOVA) followed by Bonferroni’s post hoc comparison tests. A *p*-value less than 0.05 was considered statistically significant, while a *p*-value less than 0.01 was considered highly significant.

## 3. Results

### 3.1. Detection of Cows’ Uterine Tissue

The inflamed tissues in cattle were identified through hematoxylin and eosin (HE) staining, and the apoptosis factor Caspase-8 was detected using immunohistochemistry (IHC) (Figure 2A). To assess differences in the expression of inflammatory factors, apoptosis-related factors, and antioxidant-stress-related genes between the endometritis uterus and the healthy uterus, RT-PCR was performed. The results revealed a significant increase in the expression levels of inflammatory-related factors TNF-α (*p* < 0.001), IL-8 (*p* < 0.001), and IL-10 (*p* < 0.001) in the uterus with endometritis (Figure 2B–D). Moreover, in comparison to the healthy uterus, the expression of peroxidase glutathione peroxidase (GPX) (*p* < 0.01) and catalase (CAT) (*p* < 0.01) in the uterus with endometritis was markedly reduced, and the expression of superoxide dismutase (SOD) was also lower (*p* < 0.001) (Figure 2E–G). Additionally, the levels of apoptosis-related genes FADD (*p* < 0.01) and Caspase-8 (*p* < 0.001) were significantly upregulated (Figure 2H,I).

### 3.2. Cecropin A Toxicity Assay and Concentration Screening

To assess the cytotoxicity of cecropin A on bEECs, the CCK-8 assay was employed (Figure 3). The experimental findings indicated that there was no statistically significant difference in the effects on bEEC cell activity between the 12.5 and 25 μM concentrations of cecropin A when compared to the control group. However, the 50 μM cecropin A group exhibited an inhibitory effect on bEEC cell activity after 6 h of treatment, and this inhibition became more pronounced after 12 and 24 h of treatment. From these results, it is evident that high concentrations of cecropin A can be cytotoxic to bEECs. Nonetheless, concentrations below 25 μM showed negligible cytotoxicity on bEECs. Consequently, subsequent experiments in this study were carried out using a 25 μM concentration of cecropin A.

### 3.3. Cecropin A Prevented Mitochondrial Dysfunction in bEECs

Nitric oxide, a major source of ROS in cells, showed an increased expression of NOX1 (*p* < 0.01), NOX2 (*p* < 0.001), and NOX4 (*p* < 0.001) genes in bEECs after LPS treatment compared to the untreated control (Figure 4A–C). However, these elevated expressions were significantly reduced in the cecropin A + LPS group.

Antioxidant enzymes, serving as crucial eliminators of oxygen free radicals and markers of antioxidant stress, exhibited notable changes. The expression levels of GPX (*p* < 0.01), CAT (*p* < 0.01), and SOD (*p* < 0.001) in the cecropin A + LPS group were significantly elevated compared to the LPS group (Figure 4D–F). 

The expression levels of inflammatory-related factors TNF-α (*p* < 0.001), IL-8 (*p* < 0.01), and IL-10 (*p* < 0.001) in the cecropin A + LPS group were significantly downregulated compared to the LPS group (Figure 4G–I). Furthermore, the protein expression level of the inflammatory-related factor phosphorylation-activated p38 MAPK in the cecropin A + LPS group was also significantly reduced compared to the LPS group (*p* < 0.05) (Figure 4K).

ROS and superoxide dismutase 2 (SOD2) levels in bEECs were detected, and the results are presented in Figure 5. The fluorescence intensity was significantly lower in the cecropin A + LPS-treated group compared to the LPS-treated group (Figure 5A,B). The SOD2 levels in the cecropin A + LPS-treated group were similar to those in the control group but significantly decreased compared to the LPS-treated group (Figure 5C,D).

Mitochondrial membrane potential (MMP) levels in bEECs are depicted in Figure 6. The JC-1 dye, showing red fluorescence at high levels of MMP and green fluorescence at low levels, revealed a significant increase in the cecropin A + LPS-treated group compared to the LPS-treated group. However, no notable difference was observed between the control group and the cecropin A + LPS-treated group.

### 3.4. Cecropin A Inhibited LPS-Induced Apoptosis in bEECs

Firstly, flow cytometry was employed to analyze the cell apoptosis rate (Figure 7A). The results demonstrated that the apoptosis rate of bEECs in the control group was 2.77 ± 1.11%, the apoptosis rate in the LPS treatment group was 11.57 ± 2.07%, and the apoptosis rate in the cecropin A + LPS treatment group was 7.44 ± 0.27%. Notably, the apoptosis rate in the cecropin A + LPS treatment group was significantly lower compared to the LPS treatment group (Figure 7B).

Subsequently, the expression levels of apoptosis-related proteins in the Fas/FasL-caspase-8/-3 pathway were assessed (Figure 8A). Caspase-8 (*p* < 0.01), FADD (*p* < 0.01), Caspase-3 (*p* < 0.01), and cytochrome C (*p* < 0.01) exhibited significantly lower levels in the cecropin A + LPS-treated group compared with the LPS-treated group (Figure 8B–E).

Moreover, the immunofluorescence detection of apoptotic proteins in the Fas/FasL-caspase-8/-3 pathway was conducted, revealing significantly decreased fluorescence intensity for the proteins Caspase-8 (*p* < 0.01), FADD (*p* < 0.001), Caspase-3 (*p* < 0.001), and cytochrome C (*p* < 0.001) in the cecropin A + LPS-treated group relative to the LPS-treated group (Figure 9).

The expression levels of BAX and Bcl-2 proteins related to apoptosis were also examined (Figure 10). BAX expression levels were significantly decreased (*p* < 0.01), while Bcl-2 expression levels were significantly increased (*p* < 0.05) in the cecropin A + LPS-treated group compared to those in the LPS-treated group. Moreover, the ratios of Bcl-2 to Bax were similar between the cecropin A + LPS-treated group and the control group, but both were markedly higher relative to the LPS-treated group.

## 4. Discussion

In recent years, the utilization of high-concentrate diets for feeding dairy cows has become increasingly common to achieve higher milk yields. As widely recognized, a high-concentrate diet can easily lead to an increased incidence of sub-acute ruminal acidosis (SARA) in cows, consequently resulting in an abnormally high concentration of lipopolysaccharides (LPSs) in the blood [21]. SARA-induced LPS translocation triggers endometrial inflammatory responses, resulting in damages to the endometrial epithelial barrier and physiological dysfunction, severely affecting the productivity of ruminants, and the elevated LPS levels can induce oxidative stress, trigger severe inflammatory responses, and even impact embryo implantation [24,25]. When examining the uteri of cows with endometritis, we observed oxidative stress in the endometrial tissue, accompanied by a significant increase in apoptosis factors. These findings suggest that the uterus with endometritis may not be conducive to embryo implantation. Additionally, blood tests on cows revealed a significantly higher concentration of LPS and inflammatory factors (TNF-α and IL-6) in those fed a high-concentrate diet compared to those on conventional diets. However, when cecropin A was added as a feed additive for cows on a high-concentrate diet, the levels of inflammatory factors TNF-α and IL-6 were significantly reduced. 

Antimicrobial peptides (AMPs), known for their broad-spectrum antibacterial activity, possess a strong effect on a wide range of antibacterial activities against bacteria, fungi, viruses, and other microorganisms [26]. Besides their antimicrobial action, AMPs also play roles in anti-inflammation, antioxidative stress, and anti-apoptosis in cells [27]. AMPs play a crucial role in tolerance to oxidative stress. Studies have shown that overexpressing AMP in fruit flies can resist oxidative stress by increasing antioxidase activity and preventing ROS levels from rising [28]. One study showed that a peptide called PR-39 can inhibit the apoptotic effects of several drugs with different mechanisms promoting cell apoptosis (etoposide, bleomycin, tert-butylhy droperoxide, and 2-deoxy-d-ribose) on Hela cells [29]. Cecropin A, a widely used AMP, is renowned for its remarkable anti-inflammatory and immune-boosting properties [18,20,30]. Therefore, we further investigated the mechanism by which cecropin A alleviates inflammation induced by LPS through in vitro experiments.

Studies have shown that NADPH Oxidases (NOXs) can promote ROS accumulation in cells and exacerbate inflammatory responses in certain mammalian cells, by promoting the activation of inflammatory pathways and the production of inflammatory factors [31,32]. Antioxidant enzymes, including SOD, SOD2, and peroxidases such as GPX and CAT, play a crucial role in antioxidant defense by regulating ROS levels [33,34]. A high concentration of LPS leads to abnormal ROS and MMP levels; promotes the production of inflammatory factors such as TNF-α, IL-1β, IL-6, and IL-8, causing oxidative damage and mitochondrial dysfunction [2,35]; and even leads to apoptosis [36,37]. Cecropin A has been shown to inhibit the production of TNF-α and IL-1β induced by LPS in mouse macrophage-derived RAW264.7 cells [18]. Our research revealed that LPS significantly increased the expression levels of NOX1, NOX2, and NOX4, and the addition of cecropin A significantly inhibited the upregulated expression of NOXs in LPS-treated cells. Moreover, cecropin A alleviated the reduced expression levels of SOD, GPX, and CAT in bEECs induced by LPS, thereby attenuating oxidative stress, and subsequently inhibited inflammation. Furthermore, cecropin A markedly reduced ROS and SOD2 levels while elevating MMP levels in LPS-treated bEECs. The apoptosis status of cells can be observed by commonly measuring the protein expression ratio of BAX to BCL2. Oxidative damage and the resulting mitochondrial dysfunction can decrease the Bcl-2/BAX ratio. The release of cytochrome C from the mitochondria is regulated by BAX and ultimately leads to apoptosis through the activation of Caspase-3 [38,39]. Cecropin A significantly decreased BAX levels and increased Bcl-2 levels, suggesting that bEECs exhibited superior anti-apoptotic capacity under the influence of cecropin A. It suggests that cecropin A can prevent cell apoptosis resulting from LPS-induced oxidative stress and mitochondrial dysfunction, consistent with the report that another AMP named LL-37 effectively inhibits the oxidative stress and inflammation induced by LPS [40].

The mitochondria-mediated pathway Fas/FasL-caspase-8/-3 is an important mediator of cell apoptosis through extrinsic stimulation (Figure 1) [41]. TNF-α plays a major role in inducing cell apoptosis, and LPS can stimulate the expression of inflammatory factor TNF-α. Members of the MAPK family, JNK and p38, are involved in the regulation of TNF-α secretion. JNK is mainly responsible for regulating TNF-α transcription and cell apoptosis, while p38 helps to regulate the production of TNF-α. LPS-induced macrophage apoptosis is mediated by Fas/FasL through the JNK/p38 MAPK signaling pathway [42,43]. Additionally, previous studies have shown that TNF-α and IL-1β sensitize FasL-induced Caspase-8/-3 activation and cell apoptosis by producing tBid through FasL-induced Caspase-8 activation, leading to Bax/Bak activation and release of cytochrome C into the cytosol, thereby triggering the rapid activation of Caspase-3 [6]. Furthermore, cecropin A down-regulated the mRNA expression of inflammatory factors such as TNF-α and IL-1β and displayed effective anti-apoptotic activity by inhibiting the phosphorylation of JNK and p38 MAPK [20]. Our research indicated that cecropin A reduced inflammatory factor TNF-α, IL-1β, and IL-8 levels, as well as phosphorylation-activated p38 MAPK levels, in LPS-treated cells, preventing the expression of apoptotic proteins Caspase-8, FADD, cytochrome C, and Caspase-3 in the Fas/FasL-caspase-8/-3 pathway. It can be concluded that cecropin A can inhibit LPS-induced apoptosis by mediating the Fas/FasL-caspase-8/-3 pathway.

## 5. Conclusions

An abnormally high concentration of LPS can cause oxidative stress, inflammation, and mitochondrial dysfunction and therefore activate the mitochondria-dependent apoptotic pathway Fas/FasL-caspase-8/-3, and ultimately lead to cell apoptosis, whereas this study verified that cecropin A can effectively alleviate LPS-induced oxidative stress, inflammation, and apoptosis through in vitro experiments. These results demonstrate the potential of cecropin A as a new therapy for oxidative stress and various inflammatory diseases (such as SARA), providing a solution to the problem of reduced embryo implantation rates caused by endometritis. However, single in vitro experiments cannot fully evaluate the effectiveness and feasibility of cecropin A. Pre-clinical and clinical investigations should be performed to further characterize the physiological concentrations and therapeutic potential of this product.

## Figures and Tables

**Figure 1 animals-14-00768-f001:**
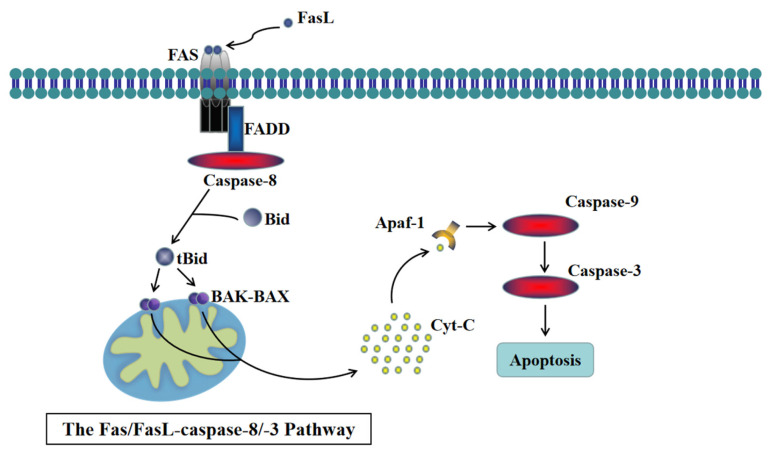
The Fas/FasL-caspase-8/-3 pathway.

**Figure 2 animals-14-00768-f002:**
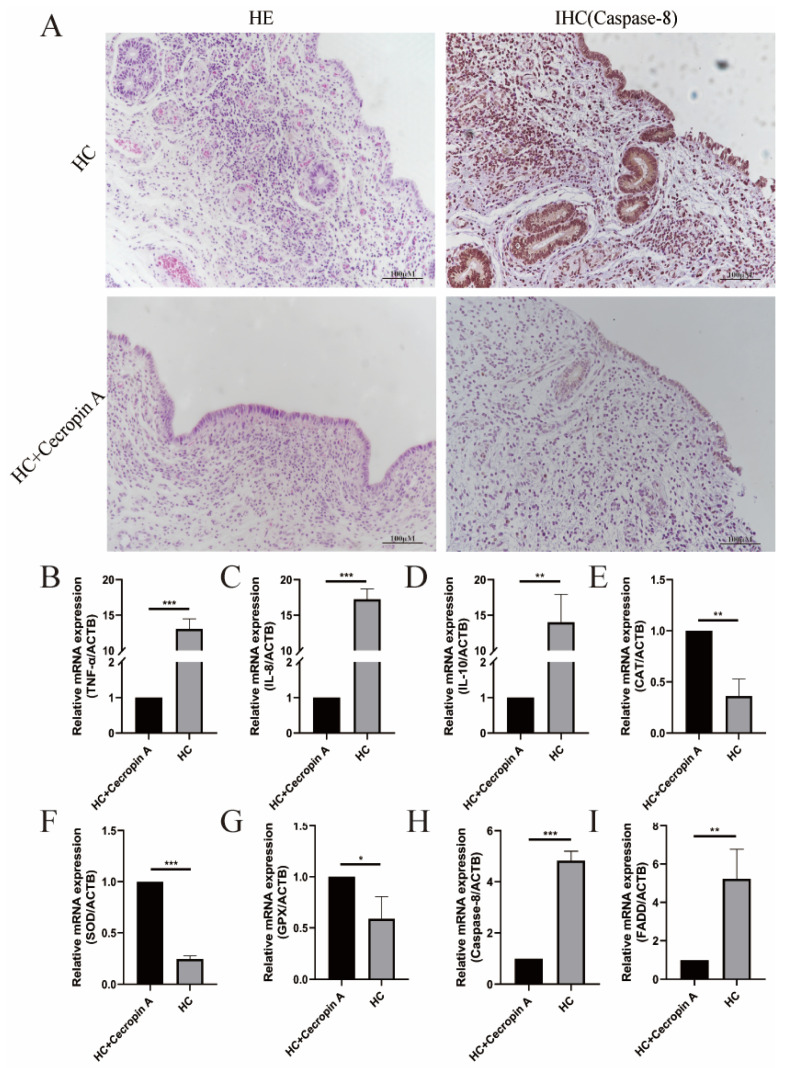
Synthesized detection of uterine tissue from cows fed with high-concentration diet (HC) and high-concentration diet supplemented with cecropin A (HC + cecropin A). (**A**) HE staining and IHC for Caspase-8 staining. (**B**–**D**) Changes in inflammation-related factors TNF-α, IL-8, and IL-10. (**E**–**G**) Changes in oxidative-stress-related genes CAT, SOD, and GPX. (**H**,**I**) Changes in apoptosis-related genes FADD and Caspase-8. * *p* < 0.05, ** *p* < 0.01, *** *p* < 0.001.

**Figure 3 animals-14-00768-f003:**
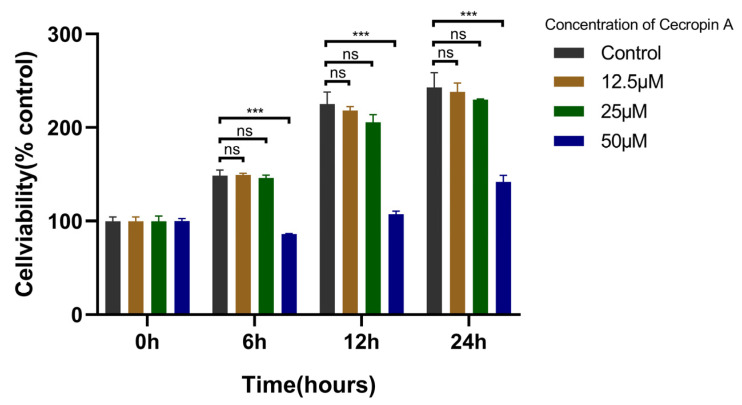
Screen of the concentration of cecropin A (0, 12.5, 25, and 50 μM) for treating bEECs by utilizing CCK-8. *** *p* < 0.001, ns *p* > 0.05.

**Figure 4 animals-14-00768-f004:**
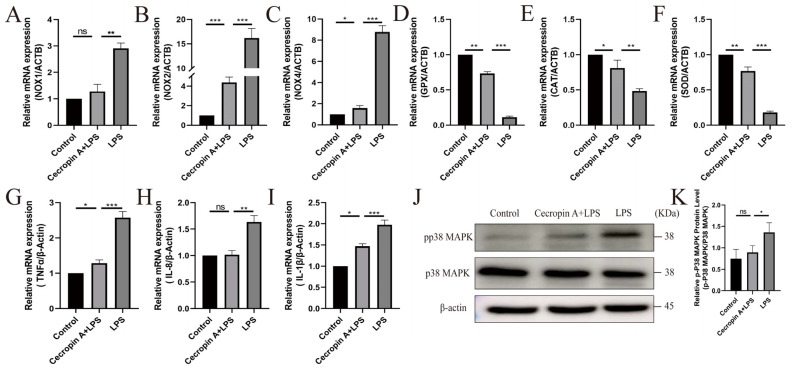
The expression levels of inflammation-related factors and oxidative-stress-related factors in bEECs. (**A**–**F**) Changes in oxidative-stress-related factors NOX1, NOX2, NOX4, GPX, CAT, SOD. (**G**–**I**) Changes in inflammation-related factors TNF-α, IL-8, and IL-10. (**J**) Western blot of the phosphorylation-activated p38 MAPK. (**K**) Relative levels of phosphorylation-activated p38 MAPK. * *p* < 0.05, ** *p* < 0.01, *** *p* < 0.001, ns *p* > 0.05.

**Figure 5 animals-14-00768-f005:**
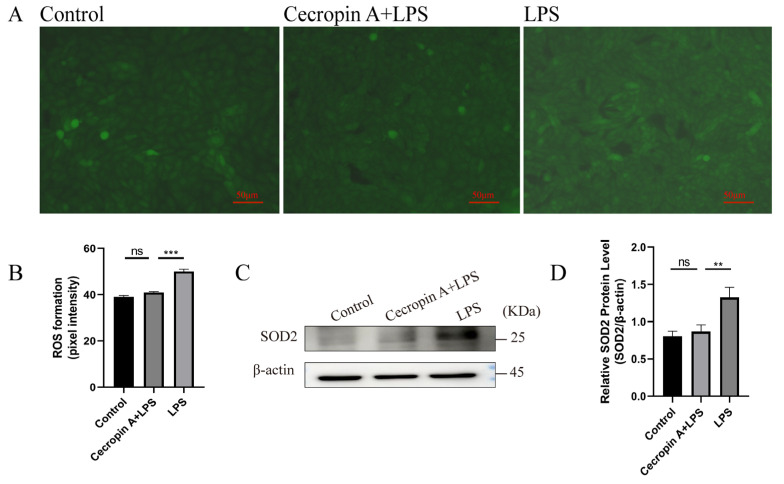
ROS and SOD2 level detection in bEECs. (**A**) Representative images of ROS levels detected using DCFH-DA fluorescence (scale bars represent 50 μm). (**B**) Quantification of intracellular ROS levels. (**C**) SOD2 level detection using Western blot. (**D**) Levels of SOD2 relative to β-actin. ** *p* < 0.01, *** *p* < 0.001, ns *p* > 0.05.

**Figure 6 animals-14-00768-f006:**
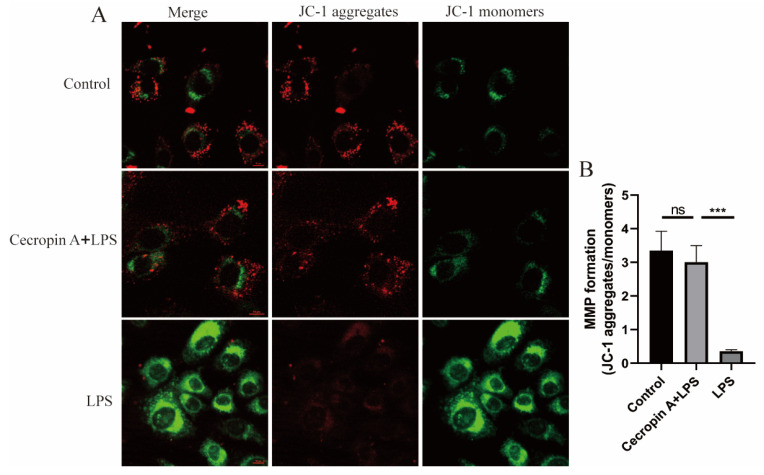
MMP level detection (scale bars represent 10 μm) in bEECs. (**A**) Representative images of MMP detection using JC-1. (**B**) Relative levels of the MMP. *** *p* < 0.001, ns *p* > 0.05.

**Figure 7 animals-14-00768-f007:**
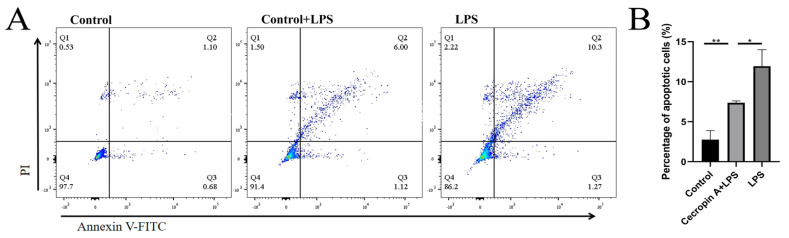
Detection of the bEECs apoptosis rate by utilizing flow cytometry. (**A**) Flow cytometry detection of the bEECs apoposis. (**B**) Relative rates of apoposis in bEECs. * *p* < 0.05, ** *p* < 0.01.

**Figure 8 animals-14-00768-f008:**
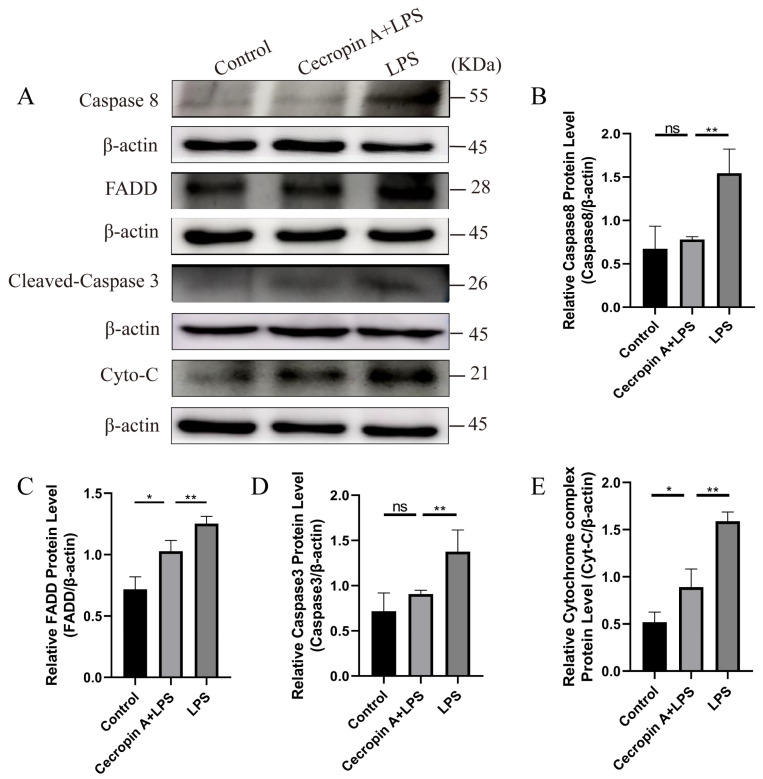
Protein levels of the apoptosis-related proteins in bEECs on the Fas/FasL-caspase-8/-3 pathway. (**A**) Western blot of the Caspase-8, FADD, Caspase-3, and cytochrome C proteins. (**B**) Relative levels of Caspase-8. (**C**) Relative levels of FADD. (**D**) Relative levels of Caspase-3. (**E**) Relative levels of cytochrome C. * *p* < 0.05, ** *p* < 0.01, ns *p* > 0.05.

**Figure 9 animals-14-00768-f009:**
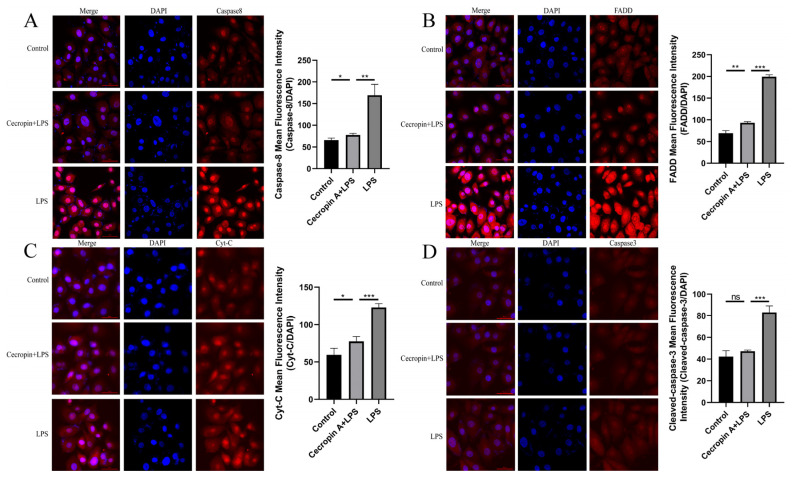
Immunofluorescence detection of apoptosis-related proteins in bEECs on the Fas/FasL-caspase-8/-3 pathway (scale bars represent 60 μm). Representative images of immunofluorescence staining and relative expression of Caspase-8 (**A**), FADD (**B**), cytochrome C (**C**), Caspase-3 (**D**). * *p* < 0.05, ** *p* < 0.01, *** *p* < 0.001, ns *p* > 0.05.

**Figure 10 animals-14-00768-f010:**
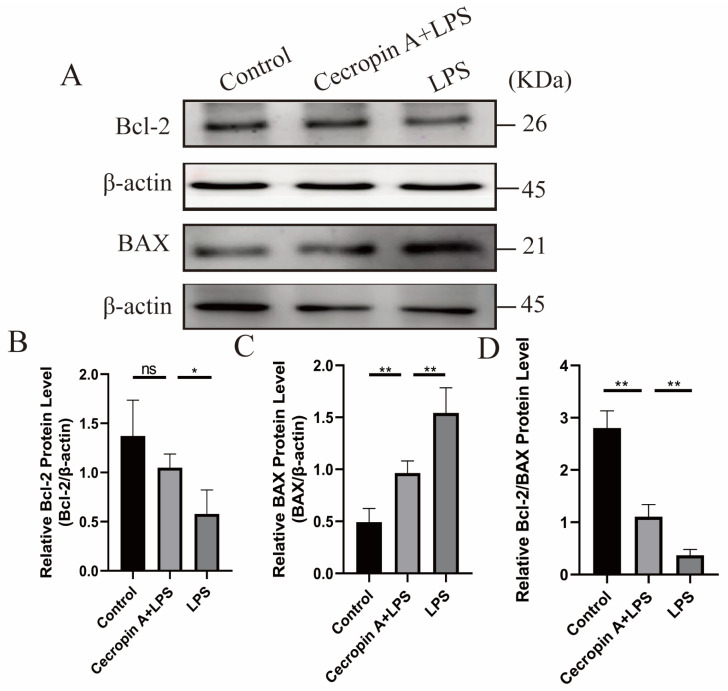
Protein levels of the Bcl-2 and BAX and Bcl-2/BAX ratios in bEECs. (**A**) Western blot of the Bcl-2 and BAX proteins. (**B**) Relative levels of Bcl-2. (**C**) Relative levels of β-actin. (**D**) The Bcl-2/BAX ratios. * *p* < 0.05, ** *p* < 0.01, ns *p* > 0.05.

**Table 1 animals-14-00768-t001:** Ingredients and nutrient contents of diets.

Ingredient (% DM)		Nutrient Composition (% DM)	
Maize	24.92	Crude protein	16.21
Soybean meal	13.48	Calcium	1.18
Barley	12.00	Phosphorus	0.51
Distiller-dried grains with solubility	5.91	Neutral detergent fiber (NDF)	29.92
Silage corn	6.00	Non-fibrous carbohydrate (NFC)	42.34
Alfalfa	17.00	Crude ash	4.87
Oat grass	17.00	Crude fat	3.05
Limestone	1.48	Starch	27.82
CaHPO_4_	0.92		
NaCl	0.37	NDF/NFC	0.71
Premix	0.92	Dry matter	48.32

**Table 2 animals-14-00768-t002:** Primer pairs used for q-PCR.

Gene Name	ID	Primer Sequences (from 50′ to 30′)	Size (bp)	Primer Efficiencies
CAT	NM_001035386.2	F: AGAGGAAACGCCTGTGTGAG R: ATGCGGGAGCCATATTCAGG	115	97.91%
SOD	NM_174615.2	F: CTCTACTTGGTTGGGGCGTC R: TCGAAGTGGATGGTGCCTTG	122	95.3%
GPx	NM_174076.3	F: AACGTAGCATCGCTCTGAGG R: GATGCCCAAACTGGTTGCAG	121	105.23%
NOX1	NM_001191340.1	F: TGTCTTTCCTGAGAGGCACC R: TTTGTGGAAGGCGAGGTTGT	80	93.22%
NOX2	NM_174035.4	F: CAAGATGGAGGTGGGCCAAT R: GAGGTCAGGGTGAAAGGGTG	81	95.44%
NOX4	NM_001304775.1	F: TCTGGACCTTTGTGCCT R: GACGGATGACTTGTGACTG	95	96.82%
TNF-α	NM_173966	F: GCTCTTACCGGAACACTTCG R: GGACACCTTGACCTCCTGAA	238	108.1%
IL-1β	NM_174093	F: AACCGAGAAGTGGTGTTCTGC R: TTGGGGTAGACTTTGGGGTCT	167	102.62%
IL-8	NM_173925.2	F: CATTCCACACCTTTCCACCC R: AGGCAGACCTCGTTTCCATT	116	93.88%
IL-10	NM_174088.1	F: CACAGGCTGAGAACCACG R: AGGGCAGAAAGCGATGA	108	95.43%
FADD	NM_001007816.1	F: CCGGAGGACCGAGACCTG R: CGTCAGATACTCCGAGGTGC	97	98.43%
Caspase-8	XM_005202615.5	F: AGCATAGCACGGAAGCAGG R: GGTCTTATCCAAAGCGTCTGC	87	96.32%
ACTB (β-Actin)	NM_173979.3	F: GGCACCCAGCACAATGAAGA R: GCCAATCCACACGGAGTACTT	67	99.4%

## Data Availability

Data available on request due to privacy/ethical restrictions (the data that support the findings of this study are available on request from the corresponding author).

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
