# Peer review of "Cecropin A Alleviates LPS-Induced Oxidative Stress and Apoptosis of Bovine Endometrial Epithelial Cells"

_animals, 2024, doi:10.3390/ani14050768_

Round 1
Reviewer 1 Report
Comments and Suggestions for Authors
This paper investigated the impact of Cecropin A on LPS-induced cell inflammation and found that Cecropin A not only achieves anti-inflammatory effects by alleviating oxidative stress but also plays an anti-apoptotic role through mitochondrial-dependent apoptotic pathways. The research results have significant reference value for solving rumen acidosis in dairy cows under high-concentrate feeding conditions. The study is very well designed, and the generated results align perfectly with the aim of the research. Except for the following minor comments, the paper is perfectly written and suitable for publication. Please correct the noted text accordingly.
1. The source of Figure one should be marked.
2. L82-85, the meaning is unclear, please rewrite this sentence and provide relevant references.
3. How can such a conclusion be drawn from L94-97? What is the rationale for this?
4. L107-108,what proportion of high concentrate were fed to these cows?
5. L120, are Bovine endometrial epithelial cells (bEECs) isolated from well-separated cell lines or fresh tissue?
6. A scale bar needs to be added to the images in Figure 2.
7. The color differences between different groups in the bar chart of Figure 3 are not obvious.
8. The color differences between different groups in the bar chart of Figure 10 are not clear.
9. L410-412, can Cecropin A solve the embryo implantation rate under endometritis? If so, it does not conform to the title and focus of this paper. That is, the conclusion cannot be drawn from the results of this paper.
10. L412-414 is unclear, please rewrite.
11. Reference 2 format is not consistent.
12. Reference 8 format is not consistent.
13. Reference 14 is incomplete.
14. The formats of references 30, 32, and 41 are need to be revised.
Comments on the Quality of English LanguageAlthough English writing is generally good, the authors need to carefully check the entire text.
Author Response
1. The source of Figure one should be marked.
Answer: We are very sorry for our careless mistake and it has been corrected in the revised manuscript (L79).
2.L82-85, the meaning is unclear, please rewrite this sentence and provide relevant references.
Answer: Thank you for your suggestion, we have modified these sentences (L82-85).
3. How can such a conclusion be drawn from L94-97? What is the rationale for this?
Answer: Previous findings of this study revealed that increased peripheral blood LPS levels caused by SARA may lead to endometritis in dairy cows. Inflammation was alleviated after feeding Cecropin A. Therefore, we validated this hypothesis through in vitro experiments.
4. L107-108,what proportion of high concentrate were fed to these cows?
Answer: Thanks for the suggestion, we have added the detailed content into the revised manuscript. (L111-113)
5. L120, are Bovine endometrial epithelial cells (bEECs) isolated from well-separated cell lines or fresh tissue?
Answer: In the revised manuscript we have added a detailed information. (L126-130)
6. A scale bar needs to be added to the images in Figure 2.
Answer: Thank you for the suggestion, we have added the scale bar to Figure 2 based on your recommendation. (L239)
7. The color differences between different groups in the bar chart of Figure 3 are not obvious.
Answer: Thank you very much for your suggestion, and we have made corresponding modifications to Figure 3. (L256)
8. The color differences between different groups in the bar chart of Figure 10 are not clear.
Answer: Thank you for your suggestion, but actually, Figure 10 is a non-color figure, and we did not pay special attention to the color. (L335)
9. L410-412, can Cecropin A solve the embryo implantation rate under endometritis? If so, it does not conform to the title and focus of this paper. That is, the conclusion cannot be drawn from the results of this paper.
Answer: Thank you for your valuable suggestion, we have provided more detailed explanations for this issue. (L413-419)
10. L412-414 is unclear, please rewrite.
Answer: We greatly appreciate your raising this issue, and we have already rewritten parts of the manuscripts accordingly. (L415-419)
11. Reference 2, 8, 14, 30, 32 and 41format is not consistent.
Answer: We appreciate your suggestion and have made a summary of questions 11-14, while also making corresponding modifications to the reference format.

Reviewer 2 Report
Comments and Suggestions for Authors
In this paper, Zahao and colleagues found that Cecropin A, relieved LPS induced oxidative stress endometrial epithelial cell and inhibited the inflammatory response caused by oxidative damage in dairy cow.
This study is very interesting and provides plenary pictures of the effect of Cecropin A, but I suggest changing some parts in Materials and methods and discussion:
2.1. Tissue Collection: Indicate the number of samples; add more information about cows (age, parity).
2.8. Cellular immunofluorescence: I suggest adding information on primary and secondary antibodies (i.e, company, country). In line 206. add more information of microscope.
And I would suggest that a quote about the negative impact of endometritis and its treatment in dairy cows be included in the discussion.
Author Response
1. 2.1. Tissue Collection: Indicate the number of samples; add more information about cows (age, parity).
Answer: Thank you very much for your suggestion. In our study, all uterine samples were collected from healthy Holstein cows (an average weight of 610kg, 1-2 parity) fed with a 100% total mixed ration (TMR), met the revised nutritional requirements (NRC) of the US National Research Council, and did not suffer from mammitis, hoof disease or other diseases underwent by veterinary clinical diagnosis. (L107-119)
2. 2.8. Cellular immunofluorescence: I suggest adding information on primary and secondary antibodies (i.e, company, country). In line 206. add more information of microscope.
Answer: Thanks for the suggestion, we have made annotations in the revised manuscript. (L205)
3. I would suggest that a quote about the negative impact of endometritis and its treatment in dairy cows be included in the discussion.
Answer: Thank you for your valuable suggestion, and we have modified the manuscript according to your suggestion. (L338-346)

Reviewer 3 Report
Comments and Suggestions for Authors
The authors assessed the effects of Cecropin A on inflammatory responses in the endometrial cells. There are a few issues that need to be addressed below.
Introduction:
The authors discuss SARA and LPS translocation from the gut in the introduction, but their present study is not designed to specifically assess the effects of SARA. Their current study is designed to look at the effects of Cecropin A on inflammation induced by LPS in endometrial cells. In cows, the LPS could be coming from many different sources, whether it be SARA or possibly an infection (metritis), or others. I'm not sure why the authors are limiting their introduction (and discussion) to just SARA, particularly because their study did not induce SARA and the cells were not collected from cows with SARA. I would recommend writing parts of the introduction a bit more broadly when discussing LPS sources and not limiting the introduction (and discussion) just to SARA. Moreover, is there any evidence in the scientific literature that LPS can translocate to the endometrium during SARA? Is there any evidence that SARA causes endometritis? Lines 92-97 need citations. Also, please provide citations that SARA reduces reproductive performance.
Materials and methods:
Tissue collection 2.1: This section is confusing, and it doesn't appear to fit with the objective of the study. I think these tissue samples were used to look at the differences in inflammation between healthy cows and cows with endometritis (data provided in Figure 2). It appears that all tissue samples came from cows supplemented with Cecropin A? Also, how did the authors determine which cows had endometritis and which cows were healthy? I am struggling to see how these data (data in Figure 2) fit with the rest of the manuscript. Isn't it common knowledge that endometritis causes inflammation? I would suggest removing these methods and data.
2.2 Cell treatment: How were the cells seeded? What concentration of cells? What type of plates were used? Do we know if the concentrations of cercropin A used are physiological?
2.3 RT-qPCR: Was primer efficiencies calculated for each primer? Please do so and report the efficiencies in Table 1. The delta delta Ct method is only valid if primer efficiencies for each primer are within 90-110%. Otherwise, please use the Pfaffl equation. Also, using 1 internal control gene (b-actin) is typically not good enough anymore. I would suggest using geNORM or other algorithms to determine if this gene was stable enough. If it is not stable enough, I would suggest adding 1 or 2 more control genes (this can be determined by geNORM).
Results:
Line 265-268: Remove. This is for the discussion.
Please check Figure 9 heading - I believe C and D are switched.
Discussion:
I would broaden this beyond just SARA (as mentioned with the introduction).
I also think you need to provide references showing that SARA induces endometritis.
Lastly, the study has limitations related to using in vitro methods. This should be discussed. Also, the concentrations used, if not physiological concentrations, should also be discussed.
Author Response
Introduction:
The authors discuss SARA and LPS translocation from the gut in the introduction, but their present study is not designed to specifically assess the effects of SARA. Their current study is designed to look at the effects of Cecropin A on inflammation induced by LPS in endometrial cells. In cows, the LPS could be coming from many different sources, whether it be SARA or possibly an infection (metritis), or others. I'm not sure why the authors are limiting their introduction (and discussion) to just SARA, particularly because their study did not induce SARA and the cells were not collected from cows with SARA. I would recommend writing parts of the introduction a bit more broadly when discussing LPS sources and not limiting the introduction (and discussion) just to SARA. Moreover, is there any evidence in the scientific literature that LPS can translocate to the endometrium during SARA? Is there any evidence that SARA causes endometritis? Lines 92-97 need citations. Also, please provide citations that SARA reduces reproductive performance.
Answer: We are very sorry for the confusion in the original manuscript. Actually, all uterine samples were collected from Holstein cows just suffering from SARA, excluding mammitis, hoof disease or any other diseases underwent by veterinary clinical diagnosis. The endometritis mentioned in the manuscript was detected after staining the uterus of the cows with SARA, and was not caused by direct infection of the uterus by pathogenic bacteria. We have modified this information in the revised manuscript (L107-119).
Materials and methods:
1. Tissue collection 2.1: This section is confusing, and it doesn't appear to fit with the objective of the study. I think these tissue samples were used to look at the differences in inflammation between healthy cows and cows with endometritis (data provided in Figure 2). It appears that all tissue samples came from cows supplemented with Cecropin A? Also, how did the authors determine which cows had endometritis and which cows were healthy? I am struggling to see how these data (data in Figure 2) fit with the rest of the manuscript. Isn't it common knowledge that endometritis causes inflammation? I would suggest removing these methods and data.
Answer: We are very sorry for the confusion for you. Actually, the uterine samples were recovered respectively from two groups of cows fed a high-quality diet (concentrate : crude = 5 : 5) alone and fed a high-quality diet supplemented with 0.012% Cecropin A. Furthermore, we are providing detailed explanations in the revised manuscript. (L111-115)
2. 2.2 Cell treatment: How were the cells seeded? What concentration of cells? What type of plates were used? Do we know if the concentrations of cercropin A used are physiological?
Answer: Thanks so much for your valuable suggestions. The bovine endometrial epithelial cells (bEECs) maintained in our laboratory, were thawed and cultured in 6-well plates in a medium comprising 90% high glucose Dulbecco's modified Eagle's medium (DMEM, BL304A, Biosharp) and 10% fetal bovine serum with 37°C in a 5% CO2 and atmospheric air environment with saturation humidity. And once the cells reached 70–80 % confluence, process them with LPS and Cecropin A. (L126-141)
3. 2.3 RT-qPCR: Was primer efficiencies calculated for each primer? Please do so and report the efficiencies in Table 1. The delta delta Ct method is only valid if primer efficiencies for each primer are within 90-110%. Otherwise, please use the Pfaffl equation. Also, using 1 internal control gene (b-actin) is typically not good enough anymore. I would suggest using geNORM or other algorithms to determine if this gene was stable enough. If it is not stable enough, I would suggest adding 1 or 2 more control genes (this can be determined by geNORM).
Answer: Thank you for your suggestions, we have added this information in the revised manuscript in Table 1 and references have been added in Line 154-155.
Results:
1. Line 265-268: Remove. This is for the discussion.
Answer: Thank you for your suggestion and we have removed these sentences.
2. Please check Figure 9 heading - I believe C and D are switched.
Answer: Thank you for pointing this out. We have modified in the revised manuscript in Figure 9. (L329-333)
Discussion:
1. I would broaden this beyond just SARA (as mentioned in the introduction). I also think you need to provide references showing that SARA induces endometritis.
Answer: Thanks for your suggestion. The cows used in our experiment were only suffering from SARA, as confirmed by veterinary clinical diagnosis and testing for the absence of other diseases. The endometritis mentioned in the manuscript was discovered after staining the uterus of the cows with ruminal acidosis and was not caused by direct infection of the uterus by pathogenic bacteria. Additionally, we have modified the revised manuscript and provided references. (L338-346)
2. The study has limitations related to using in vitro methods. This should be discussed. Also, the concentrations used, if not physiological concentrations, should also be discussed.
Answer: We are very sorry for the oversight regarding the discussion of concentration used in physiological aspects. We have modified in the revised manuscript. (Line 420-423)

Reviewer 4 Report
Comments and Suggestions for Authors
The authors present a study on the effects of cecropin on apoptosis of bovine endometrial epithelial cells.
There are some serious issues with this manuscript, which need to be addressed before acceptance. These are underlined below.
MAJOR ISSUES
-The feeding regime of cecropin must be presented in detail. Sub-section 2.1. does not present any relevant information: how much cecropin was given to animals, for how long, through which feeding regime etc. Information must be added.
-The procedure for selection of animals from which tissue samples were collected, is not shown at all.
-No control animals were used in this study. How did the authors assessed the possible beneficial effects?
The above must be clarified before possible acceptance.
MINOR ISSUE
-The objectives of the study must be clearly presented. At the moment, the relevant paragraph is vague and not well-expressed.
Author Response
1. The feeding regime of cecropin must be presented in detail. Sub-section 2.1. does not present any relevant information: how much cecropin was given to animals, for how long, through which feeding regime, etc. Information must be added.
Answer: Thank you very much for your suggestion. In our study, all uterine samples were collected from Holstein cows (an average weight of 610kg, 1-2 parity) fed with a 100% total mixed ration (TMR), and were fed at 6:00, 12:00 and 18:00 every day for 1 year. The uterine samples were recovered respectively from two groups of cows fed a high-quality diet (concentrate: crude = 5: 5) alone and fed a high-quality diet supplemented with 0.012% Cecropin A. (L107-116)
2. The procedure for the selection of animals from which tissue samples were collected, is not shown at all.
Answer: In our study, the uterine samples were recovered respectively from two groups of cows fed a high-quality diet (concentrate: crude = 5: 5) alone and fed a high-quality diet supplemented with 0.012% Cecropin A. In the high-concentrate diet feeding group, we selected cows with SARA that did not suffer from mammitis, hoof disease or other diseases underwent by veterinary clinical diagnosis for uterine tissue sampling. And the uterine samples from cows fed with a high-concentrate diet supplemented containing 0.012% of Cecropin A were sampled as control. (L111-116)
3. No control animals were used in this study. How did the authors assess the possible beneficial effects?
Answer: We are very sorry for the confusion for you. Actually, the uterine samples were recovered respectively from two groups of cows fed a high-quality diet (concentrate: crude = 5: 5) alone and fed a high-quality diet supplemented with 0.012% Cecropin A. Furthermore, we are providing detailed explanations in the revised manuscript. (L111-116)
4. The objectives of the study must be clearly presented. At the moment, the relevant paragraph is vague and not well-expressed.
Answer: Thank you for your suggestion. In this study, we delved into the detailed mechanism by which Cecropin A alleviates LPS-induced oxidative stress, inflammation, and apoptosis in bovine endometrial epithelial cells (bEECs), to confirm that Cecropin A can alleviate LPS-induced oxidative stress, inflammation, and cell apoptosis through a mitochondrial-dependent pathway, showcasing its significant potential in the treatment of various inflammatory diseases. We have modified the revised manuscript. (L26-30, L97-104)

Round 2
Reviewer 3 Report
Comments and Suggestions for Authors
Thank you for your revisions. I still think there are a few points that need to be clarified, but the paper has improved.
Section 2.1 in methods:
Because these samples are from a cow study, you must provide the full details on the cow study. We need the diet formulation (the forage to concentrate ratio isn't enough). Preferably, we would also have dry matter intakes (by treatment group) from these cows. If you cannot provide dry matter intake, then this is a major limitation that must be addressed in the discussion.
What does the ratio of concentrate : crude mean? Forage to concentrate ratio?
What was the average parity by treatment group for these cows?
How was the Cecropin A supplemented to these cows? A top dress or mixed in the feed/TMR?
Was there a "placebo" supplement for the control? What did the control cows receive? The way the paper is worded right now it appears all cows received Cecropin A.
How were treatments assigned? Was it a randomized complete block design?
Section 2.2: I am assuming that these are from a cell line and not from the cow study mentioned in Section 2.1. Please clarify.
Figure 2: I am still confused by this data. If the samples are collected from cows supplemented with Cecropin A or not then why are the groups named "endometritis" and "healthy"? Just name the groups by the treatment the cows were assigned to.
Figures 4, 5, 6, 7, 8, 9, and 10. Please provide the cell type in the figure heading. Please provide the number of samples per treatment group.
Author Response
1. Section 2.1 in methods: Because these samples are from a cow study, you must provide the full details on the cow study. We need the diet formulation (the forage to concentrate ratio isn't enough). Preferably, we would also have dry matter intakes (by treatment group) from these cows. If you cannot provide dry matter intake, then this is a major limitation that must be addressed in the discussion.
Answer: Thank you for your suggestion. We have added a table displaying the Ingredient and nutritional content of the diet in the revised manuscript.(Line124-130)
2. What does the ratio of concentrate : crud mean? Forage to concentrate ratio?
Answer: Yes, thank you for your question. We changed ''concentrate: crude'' in the manuscript to ''concentrate: forage''.
3. What was the average parity by treatment group for these cows?
Answer: Thank you for your question, we have included parity in the revised manuscript. All cows we selected for sampling were 1-2 parity. (Line107)
4. How was the Cecropin A supplemented to these cows? A top dress or mixed in the feed/TMR?
Answer: Thank you for your question. Cows as controls were fed with a high-concentrate diet supplemented with Cecropin A at a rate of 0.012% of the concentrate and then mixed in the TMR. (Line113-115)
5. Was there a "placebo" supplement for the control? What did the control cows receive? The way the paper is worded right now it appears all cows received Cecropin A.
Answer: Thanks for your question, there is no placebo supplement for the control. All cows for sampling were fed a high-concentrate diet. The difference is that only the control cows were fed with Cecropin A.
6. How were treatments assigned? Was it a randomized complete block design?
Answer: Yes, it`s a randomized complete block design. All cows in the dairy farm are fed a high-concentrate diet. Additionally, we divided some cows and added Cecropin A to their diets. For specific details, we have created a diagram to help you understand more clearly.
7. Section 2.2: I am assuming that these are from a cell line and not from the cow study mentioned in Section 2.1. Please clarify.
Answer: Yes, our experiment involves normal bovine endometrial epithelial cell lines derived from frozen in the laboratory, which are affected by inflammation simulated by LPS. It has been explained in the revised manuscript. (Line132-133)
8. Figure 2: I am still confused by this data. If the samples are collected from cows supplemented with Cecropin A or not then why are the groups named "endometritis" and "healthy"? Just name the groups by the treatment the cows were assigned to.
Answer: Thank you for your suggestion, we have adjusted Figure 2 in the revised manuscript. (Line243)
9. Figures 4, 5, 6, 7, 8, 9, and 10. Please provide the cell type in the figure heading. Please provide the number of samples per treatment group.
Answer: Thank you for your suggestion, we have provided the cell type in the headings of these figures in the revised manuscript. In addition, all values were derived from a minimum of three independent experiments for each condition in the current study. (Line 222-223)

Reviewer 4 Report
Comments and Suggestions for Authors
No further comments.
Author Response
Thank you for your suggestions, we have made the modifications to the manuscript as follows:
1. We have added a table displaying the Ingredient and nutritional content of the diet in the revised manuscript. (Line124-130)
2.We changed ''concentrate: crude'' in the manuscript to ''concentrate: forage''.
3. It has been explained the source of the bEECs in the revised manuscript for further details. (Line132-133)
4. We have adjusted Figure 2 to rename the groups in the revised manuscript. (Line243)
